# Programmatic mapping and size estimation of key populations to inform HIV programming in Tanzania

Mwita Wambura[1]ⓘ☯*, Daniel Josiah Nyato[1]ⓘ☯, Neema Makyao[2], Mary Drake[3], Evodius Kuringe[1], Caterina Casalini[3], Jacqueline Materu[1], Soori Nnko[1], Gasper Mbita[3], Amani Shao[1]ⓘ, Albert Komba[3], John Changalucha[1], Tobi Saidel[4]

**1** Department of Sexual and Reproductive Health, National Institute for Medical Research, Mwanza Centre, Mwanza, Tanzania, **2** Ministry of Health, Community Development, Gender, Elderly and Children, National AIDS Control Programme, Dar es Salaam, Tanzania, **3** Jhpiego Tanzania—an affiliate of Johns Hopkins University, Dar es Salaam, Tanzania, **4** Partnership for Epidemic Analysis (PEMA), New Delhi, India

☯ These authors contributed equally to this work.
* wmwita@yahoo.com

**Data Availability Statement:** All relevant data are within the paper and its Supporting Information files.

## Abstract

### Introduction

A programmatic mapping and size estimation study was conducted in 24 districts in 5 regions of Tanzania to estimate the size and locations of female sex workers (FSW) and men who have sex with men (MSM) to inform the HIV programming for Key Populations.

### Methodology

Data were collected at two levels: first, interviews were conducted with informants to identify venues where FSWs and MSM frequent. Secondly, the size of MSM and FSWs were estimated through interviews with FSWs, MSM and other informants at the venue. The venue estimates were aggregated to generate the ward level estimates. Correction factors were then applied to adjust for MSM/FSW counted twice or more, absent from the venues on the mapping day or remain online and hidden. The ward size estimates for mapped wards were extrapolated to non-mapped wards and aggregated to generate district and regional level estimates.

### Results

A total of 4,557 level I interviews were conducted. Further, 3,098 FSWs and 1,074 other informants at the FSWs venues and 558 MSM and 210 other informants at the MSM venues were interviewed during level II. The mapping survey identified 6,658 FSW, 1,099 FSW and MSM and 50 MSM venues in 75 wards. A total of 118,057 (range: 108,269 to 127,845) FSWs and 23,771 (range: 22,087 to 25,454) MSM were estimated in the study regions after extrapolation and accounting for correction factors. It was estimated that 5.6% and 1.3% of the female and male population of reproductive age (15–49 years old) could be FSWs and MSM in the study regions, respectively.

**Funding:** AK received a grant from the United States Agency for International Development (USAID) (SOL-621-14-000015). The funders had no role in study design, data collection and analysis, decision to publish, or preparation of the manuscript.

**Competing interests:** The authors have declared that no competing interests exist.

## Conclusion

This study provides the baseline figures for planning, target setting and monitoring of the HIV intervention services in the study areas and geographic prioritisation of the response by allocating more resources to areas with a large number of FSWs and MSM.

## Introduction

Tanzania is one of the countries with the greatest burden of HIV infection in sub-Saharan Africa (SSA) [1–3]. A nationally representative survey conducted in 2016/17 showed that the HIV prevalence among the adult population aged 15–49 years was 4.8% in Mainland Tanzania [3]. This is a decline from 5.1% and 7.0% reported in previous surveys conducted in 2011/12 and 2003/04, respectively [4]. However, there is substantial variation in HIV prevalence among different subsets of the general population, with women being more likely to be infected with HIV than men: 6.4% of women aged 15–49 years are living with HIV compared to 3.1% of men of the same age [3], and the prevalence among key population (KPs) being higher than in the general population. Data from studies conducted among KPs in selected regions of Tanzania show a prevalence of 42% among people who inject drugs [5], 28% among female sex workers (FSWs) [6] and 31.2% among men who have sex with other men (MSM) [7]. Due to their elevated levels of HIV infection, sexual risk behaviour and other practices, KPs play a critical role in the way HIV and other sexually transmitted infections spread [8]. Besides, these sub-populations face barriers to accessing general HIV prevention and care due to stigma, discrimination and criminalization [9–11]. Thus, their involvement is vital for an effective and sustainable response to HIV infection.

UNAIDS recommends that program planners use KP specific strategic information to characterise risk settings and prioritise HIV program resources [12]. Therefore, from a service/program planning perspective, it is essential that the sizes of key populations be quantified, and that locations where they can be found are identified. Various methods have been used in SSA to estimate key population size, including programmatic mapping [13–15], multiplier methods [13,14] and capture-recapture [13,16], among others. Because of the fluid nature of key populations, all studies using these methods have validity concerns and are subject to systematic biases which cannot be controlled by statistical methods or large sample sizes alone. Service multiplier methods are vulnerable to violation of independence assumptions related to data sources, as well as to sampling error, misclassification of study populations, and data quality issues in the programmatic data that are integral to their implementation, (e.g. counting people rather than visits, counting only people who are eligible to overlap with the other data sources, etc). Programmatic mapping methods are limited by their inclusion only of venue-based KP members, by challenges associated with accurately identifying KP members and by their reliance on key informants to provide the data. Capture-recapture is limited by its reliance on several assumptions that are difficult to avoid violating in the context of KP surveys, including the assumptions of independence, closed populations without in- and out-migration, and the need to accurately match people between multiple data sources [17,18]. The premise of closed communities is particularly unrealistic for KPs in Tanzania, who are reported to have high mobility [19,20].

The use of several methods of size estimation to triangulate and validate data and support the development of plausible ranges has been promoted by international agencies involved in the generation of KP size estimates such as UNAIDS and WHO [8]. Programmatic mapping

was selected for the study presented in this paper because of its ability to serve the dual purpose of providing locally relevant program data needed for determining resources required and for planning implementation strategies [21,22]. It also offers data to feed into broader regional KP size estimates [20]. Programmatic mapping was also considered more reliable for capturing a broad cross-section of KPs members, based on evidence from previous exercises in Tanzania, where comparison of results from IBBS multiplier studies in 2013 produced lower results among FSWs than mapping and enumeration studies done at the same time [2]. Use of Programmatic mapping was also justified by the understanding that most MSM and FSWs in Tanzania who are engaged in high-risk activities, solicit sexual partners by visiting physical venues. However, given the acknowledgment that this is a situation which may be rapidly changing, the study included an approach for estimating the proportion of KPs who remain hidden or do not visit physical venues.

The study was supported by the Sauti project, a 5-year project funded by the United States Agency for International Development (USAID) which aims to improve health status for all Tanzanians through a sustained reduction in new HIV infections. During this study, the Sauti project focused in 5 regions of Tanzania, including Shinyanga, Njombe, Iringa, Mbeya regions, plus Temeke district in Dar es Salaam city. The study, therefore, took place in these regions (representing 24.4% of the overall population in the country) to provide programmatic data for local program planning, target setting and monitoring of program coverage.

The programmatic mapping study intended to estimate the size and identify locations where KPs could be found in the study areas. The estimates sought to help HIV prevention programs assess their reach among KPs. Besides, the estimates sought to create an evidence base with which to direct prevention and treatment activities rationally to improve both efficiency and effectiveness. The KPs of interest were FSWs and MSM.

## Methodology

### Setting and sampling of study wards

Within the five study regions, all mapping activities took place at the ward level. The sampling strategy used to identify the study wards was based on the Tanzania administrative structure organised as five regions, 24 districts and 573 wards. The sampling frame had 573 wards which were categorised into three strata according to expected high, medium and low concentrations of KPs. The categorisation was based on a composite score. The variables included in the score were residence (urban/semi-urban/rural), population size (from 2012 census data [23] adjusted for population growth), presence of mines, plantations, harbours, truck stops, KP program interventions, and estimated number of FSW and MSM venues (from available program data). Data sources for these included published and unpublished reports from projects working with KPs, national reports, and census reports.

Due to time and resource constraints, only 75 wards could be mapped. These wards were strategically selected to maximise the likelihood of obtaining data that could be used for extrapolation in all three strata (high, medium and low). Simultaneously, it was decided to purposively select a higher number of wards in the high and medium strata (70% and 20% of study wards respectively), because of the benefit of direct mapping data to local programs. The remaining 10% of wards were selected from the low stratum. Within strata, wards were selected by systematic random sampling.

The ward was used as a sampling unit to accommodate program needs and to facilitate fieldwork planning, implementation and data collection. Previous studies conducted in sub-Saharan Africa have used smaller geographic areas called zones as units for size estimation

**PLOS** | **ONE**

[20,24]. However, in our setting, wards were preferred to zones because most informants would be aware of the venues within the ward.

## Study population

Within the selected wards, data were collected in two stages, described as Level I and Level II. Level I interviews were conducted to identify venues within the ward where FSWs and MSM go to meet new sexual partners. The study population were secondary or community informants aged 18+ years who had direct or indirect contact with MSM and FSWs and who were knowledgeable with the ward, e.g. taxi drivers, bar workers, motorcycle drivers, and health care providers serving KPs among others. The informants were selected purposively from various places such as parks, transit stops, shopping malls, nightclubs and health facilities within the ward.

During level II, interviews were conducted at the venues, with informants who were knowledgeable about the venue. The venue informants included FSWs, MSM, venue manager, owner and patrons, among others. An FSW was defined as any female who is 18+ years and who exchanges sexual activity (anal or vaginal) with a man in return for money or benefits, irrespective of the site of operation (e.g. Streets, bars, home, and hotel). An MSM was defined as any man who is 18+ years and who has sex with other men as a matter of preference or practice, regardless of their sexual identity or sexual orientation, and irrespective of whether or not they also have sex with women.

## Data collection

Before data collection in the main survey, all team members took part in the pre-test and pilot survey activities. The research team did the pre-testing by interviewing volunteers aged 18+ years from the Civil Society Organisations (CSO) working with MSM and FSWs in Dar es Salaam, Tanzania. The pilot survey was done in selected urban and rural wards of Kibaha district, in Coast region in Tanzania replicating the study procedures and populations as detailed in the protocol. The pre-test and pilot survey activities aimed to test the appropriateness of the research procedures and data collection tools and to assess the feasibility of the work plan, staffing and other activities in the research team.

Data collection for main survey took place between April and September 2016, starting with preliminary visits to the regional and district headquarters to explain the study rationale, objectives and procedures to the authorities. National, regional and ward stakeholder meetings were convened to discuss procedures that guarantee the safety of the research team and KPs from social harm events, the confidentiality of the study participants and objectives of the study.

Data collection during level I began as a casual conversation with informants to build rapport and gather information discreetly about venues that KPs visit to meet new sexual partners. For each venue mentioned, information that characterised the venue was collected (public place, brothel and nightclubs), hours of operation, and the minimum and maximum estimated number of KP members who could be found at the venue at different times (peak time on a peak day, during the week). Venues were categorised as fixed sex venues (spots where clients visit to initiate sexual contact with FSW or MSM, e.g. brothels, hotels, guest houses or other locations where sexual activities can take place) and entertainment venues (spots where FSWs and MSM go for entertainment and soliciting sexual partners, e.g. bars, nightclubs, spa and massage parlours). Other categorisation included social venues (spots where MSM and FSWs go for social activities and partner solicitation, e.g. malls, shopping centres, markets, food stalls and fitness centres). Street venues included streets, bus and truck stop areas, park, and harbour

areas while other venues were defined as other types of venues used by MSM and FSWs for partner solicitation but not classified above, e.g. educational facilities. About 60 interviews were conducted with informants in the study ward to identify venues frequented by FSWs and MSM using a pre-designed format. This sample size per ward was considered sufficient to achieve saturation on the identification of venues frequented by FSWs and MSM within the ward.

Following the daily Level I activity, data were assembled and reviewed every day, followed by sorting into various venue lists, which served as a foundation for the next level of activity. A tablet was used to collect data while a computer database was used for data collation and analysis. The primary outcome of this phase was to develop level I unique spot list/ Level I master list. Before level II data collection, all venues were visited to validate their existence and to obtain permission from community gatekeepers (venue owners and managers) to initiate dialogue with level II informants at the venue. This phase also involved assessing venue particulars, e.g., whether the venue is safe for the interviews to be done using tablets or availability of private and suitable locations for respondent interviews as well as general safety planning. This visit was also used to update the level I master list (removing inactive venues and duplicated venues) from the master list.

During level II, data collection activities involved conducting interviews with informants at the venue using local MSM and FSWs as mobilisers and interviewers. Each regional team was comprised of a post-graduate epidemiologist who was also the team leader and the regional coordinator. The team leader was assisted by 5–9 supervisors who managed a fieldwork team of interviewers sub-divided into 5–9 MSM and 5–9 FSW sub-teams depending on the number of study wards in the region. Each MSM sub-team comprised of 4 interviewers (including 2 MSM) and two mobilisers while each FSW sub-team comprised of 4 interviewers (including 2 FSWs) and two mobilisers. The supervisors had university education while all interviewers had at least form six level of schooling with age ranging from 20 to 45 years and were recruited in the study regions.

Eligible informants for venues attended by MSM were MSM and other informants at the venue, aged 18+ years, willing and able to give oral informed consent. Similarly, eligible informants for FSW venues were FSWs and other informants at the venue, aged 18+ years, willing and able to give oral informed consent. Level II informants were asked to estimate the minimum and the maximum number of MSM/FSWs visiting a venue during a typical day, peak day and peak time and typical week using a pre-designed format. Level II estimates were used to generate final estimates. Level II activities also entailed collecting data to correct for the less visible segments of the population as much as possible. Correction factors were developed for double counting–to adjust for people who were counted twice or more at different venues (overlap between various venues in the ward). An adjustment was done for the frequency of visiting venues to inflate for the proportion of people who visit venues less frequently, e.g. less than once a week; and, invisibility to inflate for people who are absent from the venues included in the study because they find partners exclusively from the internet or through friends. Table 1 details the questions that were asked to collect data for correction factors.

**Quality assurance.** The team leaders, supervisors and interviewers were responsible for all aspects of the quality and consistency of data. To ensure that high-quality data was collected, morning meetings were conducted daily with the field staff. During these meetings, daily plans were discussed with the team, and feedback about the quality of data collected on the previous day was provided. Data queries were generated by the NIMR Mwanza data management unit within 24 hours and sent to the team in the field. All issues faced in the field were discussed with the supervisors and the team leaders. Team leaders and supervisors also documented any social harm event that happened, severity, action taken, approximate onset and resolution dates.

**Table 1. Questions used to derive the adjustment factors for FSWs.**

| Type of Adjustment factor | Questions |
|---|---|
| **Double Counting adjusted estimate** | **In the last seven days, including today, how many FSW work/visit here to solicit sexual partners (min-max)?**<br>*Minimum:* \|___\|___\|___\|   *Maximum:* \|___\|___\|___\|   *Don't Know:* \|___\|<br>*No Response:* \|___\| |
| | **Of the FSWs who came here to solicit sexual partners at least once per week, how many FSWs visited other venues to meet clients in the last seven days?**<br>*Minimum:* \|___\|___\|___\|   *Maximum:* \|___\|___\|___\|   *Don't Know:* \|___\|<br>*No Response:* \|___\| |
| | **Of the FSWs who came here to solicit at least once per week, how many different places on average do they go to meet clients in the last seven days, including this place?**<br>*Minimum:* \|___\|___\|___\|   *Maximum:* \|___\|___\|___\|   *Don't Know:* \|___\|<br>*No Response:* \|___\| |
| **Frequency of visiting venues adjusted estimate** | **Of the FSWs who came here to solicit for sexual partners at least once per week, how many FSWs come here (please read the question and categories slowly)**<br>*At least once per week* \|___\|___\|___\|<br>*Less than once per week but more than once per month* \|___\|___\|___\|<br>*At least once per month* \|___\|___\|___\|<br>*Less than once per month but more than once per 3 months* \|___\|___\|___\|<br>*Did not come here in the last 3 months* \|___\|___\|___\| |
| **Hidden/invisible population-adjusted estimate** | **How many FSWs do you know or associate within this ward**<br>*Estimate:* \|___\|___\|___\|   *Don't Know:* \|___\|   *No Response:* \|___\| |
| | **How many of these FSWs you know in this ward do not EVER come to any venue to pick clients.**<br>*Estimate:* \|___\|___\|___\|   *Don't Know:* \|___\|   *No Response:* \|___\| |

A five-day training of team leaders and supervisors was done to improve the quality of data collected. The team was trained on the manual of operations (MOP) and study protocol, interviewing techniques, research ethics and strategies to improve validity, reliability and completeness of the data collected. The interviewers and mobilisers where trained by team leaders and supervisors in the study regions using a training package provided and supervised by investigators.

Random spot checks were done by investigators to ensure the validity and quality of information collected. The fieldwork was also supervised by investigators who conducted monitoring and quality assurance activities following an approved manual of operations and protocol. Investigators also monitored social harm events to ensure that they were adequately resolved and addressed security issues and data collection challenges. Field supervision was provided continuously throughout the data collection period.

To safeguard the safety of the KP communities, first, the study team ensured that KP community leadership was actively involved in all procedures of the study. Any concern expressed by the KP community members was addressed to their satisfaction. Second, all participants were recruited into the study after a detailed description of study procedures and obtaining informed consent. All interviews were done in private areas, and interviewers were trained to ensure that the initial contact did not compromise the safety of KPs. Third, a non-identifying coding system was used to track study data, and all survey materials were kept in securely locked cabinets. Fourth, meetings were done with stakeholders, including law enforcing agencies to explain the objectives of the study, timing and all stakeholders, including law enforcing agencies, provided support where needed.

**Statistical analysis.** A database with inbuilt quality checks was developed in open data kit. The dataset was managed and analysed in Stata 13.1 (STATA Corp 2016). Venue estimates

were generated by averaging the mean/minimum/maximum informant estimates across informants interviewed in a venue. Ward level unadjusted estimates were obtained by summing across venue estimates in the ward. Data for adjusting for double counting of MSM/FSW between venues in the ward was collected from the FSWs/MSM interviewed at each venue during the survey. The formula [25] that was applied to correct for MSM/FSWs who visit multiple venues is given below:

$$Ward\ Level\ double\ counting\ adjusted\ estimate\ (WDSE) = N - (N * P * (1 - \frac{1}{S})) \qquad (1)$$

Where N = Number of MSM/FSWs estimated to have visited each venue in a week, summed across all venues in the ward. P = percent of MSM/FSWs double-counted in a week, averaged across all venues in the ward, S = average number of venues visited by MSM or FSWs for a week.

Frequency correction factors were also collected to inflate for KPs who visit the venues less frequently (less than once per week). Activities at the venue vary on different days of the week, and MSM and FSWs may frequent more to some sites than others during the week of data collection. To adjust for the frequency of visits in the venues, the formula below was applied[25]:

$$\mathbf{WDFS} = (WDSE * percentweekly * 1) + (WDSE * percentbi - weekly * 2) + (WDSE \\ * percentmonthly * 4.3) \qquad (2)$$

Where: 1) WDFS = Ward level estimate adjusted for double-counting and frequency and 2) WDSE = ward level estimate adjusted for double-counting. The values of 1, 2 and 4.3 were chosen to represent the weight of "1" for people who visit venues every week on average, the weight of 2 for people who visit bi-monthly (once every two weeks) on average, and the weight of 4.3 for people who visit monthly (once every 4.3 weeks) or less on average.

There are some MSM/FSWs who exclusively use the internet, mobile-based applications and friends to find sexual partners. This subset of KP members is difficult to account for in programmatic mapping studies. However, we attempted to account for them by including an additional adjustment factor which provided a rough estimate of the percentage of KPs who do not appear at physical venues. The formula [25] for this adjustment is provided below:

$$\mathbf{WDFHS} = \frac{WDFS}{(1 - Percent\ hidden\ ward\ correction\ factor)} \qquad (3)$$

Where WDFHS = Ward double-counting, Frequency and Hidden Adjusted Size, WDFS = Ward double-counting and Frequency Adjusted number.

The final estimates adjusted for three different correction factors at ward level were presented as ranges with minimum and maximum estimates. Ward level point estimates were calculated by averaging the minimum and maximum estimates. For all mapped wards, the adjusted estimates were divided by the size of the male and female population aged 15–49 years in the ward using 2012 census data (projected forward to 2016), to obtain population proportions. Using the programmatic mapping estimates, we calculated a proportion of women and men of reproductive age (15–49 years) who could be FSWs/MSM in each ward included in the study.

**Extrapolation.** First, linear regression modelling was used for estimating the size of FSWs/MSM in the wards without direct estimates within the study regions. The adjusted size estimate of the FSW/MSM in the wards which were directly mapped was used as an outcome variable while the male/female population aged 15–49 years in the ward and strata were used as covariates. The strata was a variable developed using cut-points from the composite score

and was used in the selection of the study wards. It was assumed that there is a relationship between the FSW/MSM population estimate and the covariates included in the model. However, the regression models resulted in many negative values, especially for MSM and larger values than those directly mapped for FSWs, which reflected the imprecision of the covariates in predicting the outcome variable.

Stratified imputation was then used for estimating the size of FSWs/MSM in the wards without direct estimates within the study regions. All wards (with and without direct estimate) in the study regions were re-stratified post-mapping into three strata (high, medium and low). The re-stratification was done after observing that the expected relationship between the key population sizes and the criteria used to develop the original strata did not hold. This happened mainly because of push and pull factors between wards which distorted the relationship between the number of KPs in the ward and the size of the population residing in the ward. Such pull and push factors included the presence of socio-economic activities that attract males as clients of sex workers, and consequently, females who sell sex. The purpose of post-mapping stratification was to group wards into strata that were more likely to have similar population proportions of MSM/FSWs.

The mean value of stratum-specific population proportions was computed for the enumerated wards in the strata within the study region. The mean stratum-specific population proportion was imputed to non-enumerated wards in the strata within the region. Mapped and extrapolated estimates were then aggregated to generate estimates of FSW/MSM at the district and regional level.

**Validation of data collected.** The programmatic mapping study included an initial listing of MSM/FSWs venues during level I and visiting these venues to verify if they are still functional and interviewing informants at the venue during level II. This approach allowed validation of the venues identified during level I.

In wards that participated in programmatic mapping study and had programs providing HIV services among FSWs and MSM, a comparison of the programmatic mapping study estimate and the 12-months HIV program service delivery data was done to validate study estimates. However, the definitions used for FSWs in the study and HIV programs were somewhat different. FSWs receiving Sauti HIV intervention services were defined as female adults aged 18+ years who receive money, goods or favours in exchange for sexual services as a primary source of income, either regularly or occasionally [26]. MSM were defined as men who engage in sexual relations with other men regardless of the motivation.

The extrapolated mapping estimates were compared to the previous estimates generated in the study regions. In 2014, key population stakeholders in Tanzania used a Delphi method to seek consensus on the estimated size of FSWs, MSM and people who use/inject drugs (PWUD/ PWID) across all regions in Tanzania. Finally, the mapping estimates were shared in various forums with key population civil society organisations, project staff providing HIV services and other stakeholders to get their expert opinion. Where there was a considerable discrepancy between HIV service delivery data and mapping estimate, an explanation was sought.

## Ethics

The study was granted ethics approval by the Medical Research Coordinating Committee (MRCC) of the National Institute for Medical Research (NIMR) in Tanzania (IRB00002514; FWA00002632) with a letter with reference number NIMR/HQ/R.8a/Vol.IX/2086 and Institutional Review Board of the Johns Hopkins University (IRB 00006668, FWA00000287).

Eligible informants were males and females aged 18+ years and willing and able to give oral informed consent. Interviewees provided verbal consent because sex work and same-sex

relationships are illegal in Tanzania. The interviewers signed the consent form to confirm that they had obtained informed consent from the participant before conducting the interview. During the informed consent process, respondents were informed that participation was voluntary and that non-participation had no negative consequences in terms of access to programs or services. They were also informed that the study did not pose risks to them, and the data collected would be kept strictly confidential and used for research purposes only. No personal identifying information was collected from respondents as part of the study.

## Results

A total of 4,557 interviews were conducted with key informants during level I to identify 7,807 venues (6,658 FSW, 1,099 FSW and MSM, and 50 MSM venues) in the 75 study wards. Of the informants, 803 (17.6%) were aged 18–24 years, 2,566 (56.3%) were aged 25–34 years, and 1,188 (26.1%) were aged 35+ years. During level 2, 3,098 (74.3%) FSW and 1,074 (25.7%) other informants at the FSWs' venues and 558 (72.7%) MSM and 210 (27.3%) other informants at the MSM venues were interviewed to estimate the size of FSW and MSM population at the venue.

Of the 1149 MSM venues identified, 726 (63.2%) were in Temeke district. The FSWs venues (n = 7757) were spread across the study regions with 2847 (36.7%) in Mbeya region, 1381 (17.8%) in Temeke district, 1249 (16.1%) in Iringa region, 729 (9.4%) in Njombe region and 1551 (20.0%) in Shinyanga region. Across all study regions, entertainment venues were the most frequented venues by FSWs 6043 (77.9%) and MSM 824 (71.7%).

### Size estimates in mapped wards and application of correction factors

An estimated population of FSWs in 75 mapped wards after correcting for double-counting was 7,505 (range: 5,847 to 9,163) and 769 (range: 567 to 970) MSM. Table 2 illustrates the calculations of adjustment factors for FSW mapped estimates for Yombo Vituka ward in Temeke district. When the correction factor for double-counting was applied, the FSW estimate decreased from 194 to 123, about 36.6% reduction. The FSW estimate for Yombo Vituka ward increased from 123 to 204 after correcting for the frequency of visiting venues, a 65.9% increase. The final estimate increased to 217 after adjusting for the invisible population. Thus, the final estimate after applying the three correction factors in the ward is 217. Similar adjustments were made for MSM mapped estimates.

The mobility adjusted number of FSWs and MSM was further analysed by the type of venue from where the FSWs and MSM operated. Across all five study areas, most of the FSWs (75.4%) and MSM (82.0%) were based at entertainment venues (S1 Fig and S2 Fig). The percentage of FSWs at fixed sex venues as a percentage of all FSWs in the region was high in Temeke (35.1%), Shinyanga (26.5%) and 19.7% in the Njombe region.

### Extrapolated regional per capita estimates

The size estimates adjusted for double-counting in the study areas was 43,897 (range 37,041 to 50,752) among FSWs and 5,025 (range 4,378 to 5,671) among MSM across 5 regions. The estimates increased to 81,445 (range 72,509 to 90,381) FSWs and 15,080 (range 12,951 to 17,208) MSM after inflating for frequency of visiting venues. This is almost 85.5% and 200.1% increment among FSW and MSM, respectively. These estimates were inflated further to 118,057 (range 108,269 to 127,845) FSWs and 23,771 (range 22,087 to 25,454) MSM to account for MSM/FSWs who do not visit the venues (hidden population). The invisibility correction factor increased the estimate by 45.0% among FSWs and 57.6% among MSM. The estimated number

**Table 2. Applications of Adjustment Factors among FSWs in Yombo Vituka Ward in Temeke District.**

| Double Counting Adjustment | Number or Percent of FSWs at Yombo Vituka Ward |
|---|---|
| Unadjusted number of FSWs observed in the ward during the week (summed across all venues in the ward) | 194 |
| Proportion estimated to solicit at 2 or more venues during the week | 44.0% |
| The average number of venues visited by FSWs over the course of the week | 6 |
| The double counting adjusted estimate among FSWs | 194 −(194*0.44)* (1-1/6) = 123 |
| **Frequency Adjustment** | |
| Double counting adjusted estimate of FSWs in the ward during the week | 123 |
| The estimated proportion of FSWs who visit the venues at least once per week (as a % of all FSWs who solicit at venues) | 72.4% |
| The estimated proportion of FSWs who visit the venues once bi-weekly (as a % of all FSWs who solicit at venues) | 10.8% |
| The estimated proportion of FSWs who visit the venues once monthly (as a % of all FSWs who solicit at venues) | 16.8% |
| The calculation for frequency adjustment for FSWs who were not present at enumerated sites during the week | (123*0.724*1) + (123*0.108*2) + (123*0.168*4.3) = 204 |
| **Hidden Population Adjustment** | |
| Double counting and Frequency adjusted estimate of FSWs in the ward during the week when mapping occurred | 209 |
| Estimated invisible/hidden proportion (as a % of all FSWs who solicit in the ward) | 3.7% |
| FSW size estimate with all three adjustment factors (double-counting, frequency of visiting venues and hidden portion of the population) applied | 209/(1−0.037) = 217 |

of FSWs and MSM in the study regions after adjusting for correction factors is shown in Tables 3 and 4.

The 2012 census data projected to 2016 and disaggregated by gender was used to estimate the proportion of women and men of reproductive age (15–49 years) who could be FSWs/ MSM in each of the regions. The per capita FSWs was 7.7% (range: 7.0%-8.4%) in Iringa region, 8.1% (6.7%-9.5%) in Njombe and 6.8% (6.4%-7.2%) in Mbeya region, 5.7% (5.4%-5.9%) in Shinyanga region and 2.1% (1.9%-2.3%) in Temeke district. The combined per capita FSW across the study regions was 5.6% (range: 5.2% to 6.1%). Conversely, the per capita MSM was 2.9% (range: 2.8%-2.9%) in Iringa region, 0.2% (0.2%-0.2%) in Njombe and 0.5% (0.5%-

**Table 3. Extrapolated FSWs within study region.**

| | Temeke | Mbeya | Njombe | Shinyanga | Iringa | Total |
|---|---|---|---|---|---|---|
| Females aged 15–49 years in 2016 | 528,797 | 761,840 | 181,614 | 382,217 | 241,090 | 2,095,558 |
| **FSW estimate adjusted for Double Counting (DC)** | | | | | | |
| Range in absolute numbers | 3758; 7947 | 16,192; 20,463 | 5052; 5971 | 3849; 7386 | 8190; 8985 | 37,041; 50,752 |
| % of FSWs aged 15–49 years | 0.7%; 1.5% | 2.1%; 2.7% | 2.8%; 3.3% | 1.0%; 1.9% | 3.4%; 3.7% | 1.8%; 2.4% |
| **FSW estimate adjusted for DC and Frequency** | | | | | | |
| Range in absolute numbers | 7598; 9512 | 34,965; 41,026 | 8270; 11,062 | 9832; 13678 | 11844; 15103 | 72,509; 90,381 |
| % of FSWs aged 15–49 years | 1.4%; 1.8% | 4.6%; 5.4% | 4.6%; 6.1% | 2.6%; 3.6% | 4.9%; 6.3% | 3.5%; 4.3% |
| **FSW estimate adjusted for DC, Frequency and invisibility** | | | | | | |
| Range in absolute numbers | 10,124; 12,493 | 48,457; 54,995 | 12,142;17,333 | 20,766, 22,665 | 16780; 20359 | 108,269; 127,845 |
| % of FSWs aged 15–49 years | 1.9%; 2.4% | 6.4%; 7.2% | 6.7%; 9.5% | 5.4%; 5.9% | 7.0%; 8.4% | 5.2%; 6.1% |

**Table 4. Extrapolated MSM within the study region.**

| | Temeke | Mbeya | Njombe | Shinyanga | Iringa | Total |
|---|---|---|---|---|---|---|
| Males aged 15–49 years | 498,499 | 661,419 | 153,678 | 348,130 | 219,478 | 1,881,204 |
| **MSM estimates adjusted for Double Counting (DC)** | | | | | | |
| Range in absolute numbers | 1823; 2191 | 248; 399 | 96; 114 | 1123; 1492 | 1088; 1475 | 4378; 5671 |
| % of MSM aged 15–49 years | 0.4%; 0.4% | 0.0; 0.1% | 0.1%; 0.1% | 0.3%; 0.4% | 0.5%; 0.7% | 0.2%; 0.3% |
| **MSM estimates adjusted for DC and Frequency** | | | | | | |
| Range in absolute numbers | 3739; 6741 | 1111; 1159 | 362; 402 | 2492; 3393 | 5247; 5513 | 12951; 17208 |
| % of MSM aged 15–49 years | 0.8%; 1.4% | 0.2%; 0.2% | 0.2%; 0.2% | 0.7%; 1.0% | 2.4%; 2.5% | 0.6%; 0.8% |
| **MSM estimates adjusted for DC, Frequency and invisibility** | | | | | | |
| Range in absolute numbers | 6441; 8815 | 3038; 3053 | 406; 432 | 6018; 6713 | 6184; 6441 | 22087; 25454 |
| % of MSM aged 15–49 years | 1.3%; 1.8% | 0.5%; 0.5% | 0.2%; 0.2% | 1.7%; 1.9% | 2.8%; 2.9% | 1.0%; 1.2% |

0.5%) in Mbeya region, 1.8% (1.7%-1.9%) in Shinyanga region and 1.5% (1.3%-1.8%) in Temeke district. The combined per capita MSM across the 5 study regions was 1.3% (range: 1.2% to 1.4%).

## Validation of the mapping estimates

The size of FSW and MSM population who visit venues (a reachable segment of the FSW/MSM population) obtained through the programmatic mapping were compared to the 12-months HIV program service delivery data collected through the Sauti project. Table 5 presents the comparison between Sauti project service uptake and directly mapped estimates among FSWs and MSM in the wards that participated in the programmatic mapping size estimation and received KP interventions. Except in Iringa region where the double counting and frequency estimate of the FSWs was higher than the number of FSWs who were reached by

**Table 5. Comparison between Sauti project service uptake and directly mapped estimates.**

| Region/District | HIV program service delivery data § | | Mapping Estimate € (Range in absolute number) | |
|---|---|---|---|---|
| | Accessed behaviour change education ⸹ | Accessed Biomedical services | Low | High |
| **FSWs** | | | | |
| Temeke | 3007 | 4349 | 2349 | 3698 |
| Mbeya | 977 | 3433 | 3320 | 4940 |
| Njombe | 1134 | 2231 | 860 | 1323 |
| Shinyanga | 2045 | 935 | 941 | 1788 |
| Iringa | 1037 | 1033 | 2108 | 3234 |
| **MSM** | | | | |
| Temeke | 1442 | 932 | 1103 | 1802 |
| Mbeya | 88 | 178 | 10 | 28 |
| Njombe | 187 | 172 | 6 | 13 |
| Shinyanga | 285 | 13 | 194 | 441 |
| Iringa | 28 | 26 | 18 | 32 |

§ Annualised estimate of FSWs and MSM. For FSWs, only FSWs who are 18+ years and with at least 50% of income coming from sex work are included in the HIV program data.

⸹ Number of FSWs/MSM reached with individual and/or small group-level behavioural HIV prevention interventions

€ Mapping estimate adjusted for double counting and frequency

**Table 6. FSWs mapping size estimates vs previous studies.**

| Region/District | Consensus report produced in 2014 | IBBS done in 2013 | Mapping (Double Counting adjusted Estimate) | Mapping (Double Counting and Frequency adjusted Estimate) |
|---|---|---|---|---|
| FSWs † | | | | |
| Dar es Salaam | 22,500; 34,645 | 3,502; 8,031 | 3758; 7947 ¶ | 7598; 9512 ¶ |
| Mbeya | 9187; 12,000 | 3928; 11,838 | 16,192; 20,463 | 34,965; 41,026 |
| Njombe | 2981; 6000 | NA | 5052; 5971 | 8270; 11,062 |
| Iringa | 4000; 8000 | 2060; 4009 | 8190; 8985 | 11844; 15103 |
| Shinyanga | 6000; 9000 | 3541; 6306 | 3849; 7386 | 9832; 13678 |

† No estimates for MSM at the regional level. Few studies were done in Dar es Salaam out of the 5 study regions

NA—IBBS was not done in Njombe region

¶ Estimates for Temeke district only, one district out of three districts in Dar es Salaam in 2014.

Sauti project interventions, the mapping estimates are consistent to Sauti programmatic reach among FSWs in other regions. Among MSM, the number reached by the Sauti project interventions was high in Mbeya and Njombe regions while the mapping estimates in other regions were consistent to Sauti programmatic reach.

The size of FSW and MSM population who visit venues (double-counting adjusted estimate) obtained through the programmatic mapping were also compared to previous estimates. The mapping estimates were compared to the consensus report generated by the National AIDS Control Programme (NACP) in 2014 through a Delphi based process and the IBBS data collected in 2013 [2]. The IBBS used respondent-driven sampling (RDS) to recruit FSWs in the study. The double-counting adjusted estimate for FSWs in Iringa and Mbeya regions were higher than the consensus report estimates and integrated bio-behavioural survey estimates (Table 6). The double-counting adjusted estimates for FSWs were consistent to estimates reported in the consensus report for Njombe region and Shinyanga region. In Dar es Salaam region, only one district (Temeke) out of three districts was mapped. The double-counting and frequency adjusted estimate for FSWs in all study regions were higher than the consensus report estimates and integrated bio-behavioural survey estimates except in the Dar es Salaam region.

## Discussion

This was the first large scale study to physically map the location and estimate the sizes of FSWs and MSM subpopulations across the five study regions of Tanzania. Overall, it was estimated that 118,057 (range 108,269 to 127,845) FSWs which represents 5.6% (range: 5.2% to 6.1%) of the female population aged 15–49 years in the study regions were FSWs. Similar estimates have been obtained in a mapping size estimation study in Kenya which reported that 5% of the female population of reproductive age (15–49 years) of the urban female population are FSWs [27]. However, the figure reported in the Kenyan study is adjusted for only double counting. These estimates may be lower than findings from previous studies done elsewhere in sub-Saharan Africa among FSWs. For instance, a study conducted in a provincial town in Madagascar in 2001 using the capture-recapture method showed that 12% of the female population aged 15–49 years in that town were FSWs [28,29].

Similarly, previous studies conducted in Tanzania estimated that the population of MSM in Dar es Salaam ranged from 6,409 to 13,513 [30] and 3,703 in Dodoma region [31]. In the current study, we estimated the MSM population at 23,771 (range 22,087 to 25,454) MSM representing 1.3% (range: 1.2% to 1.4%) of the male population aged 15–49 years in the study

regions were MSM. This percentage of males who are aged 15–49 years who are MSM in this study is consistent with the estimate reported in the consensus report, which is 1.7% (range: 1.4–2.4)[2].

The southern highland regions of Iringa, Mbeya and Njombe, had the highest percentage of the female population aged 15–49 years who are FSWs while Shinyanga and Iringa regions and Temeke district had the highest percentage male population who are MSM. This suggests that these areas have factors that pull a large number of FSWs and MSM. The Southern Highland regions have a large number of truck drivers plying along the Dar es Salaam to Zambia transport corridor, tea plantations, timber production and the highest HIV prevalence in the country. Temeke is one of the districts in Dar es Salaam city, and Shinyanga region has several large and small-scale gold mines. Previous studies have shown that "sex workers follow the money" by migrating to areas with a large number of men who can pay for sexual services [32,33]. The large per capita FSW population in the Southern Highland regions represent high HIV epidemic potential and justifies continued targeted prevention efforts among KPs to sufficiently saturate the regions with HIV interventions.

In this study, three adjustments factors were applied. Estimates corrected for double-counting represents KPs who visit the venues at least once per week and therefore a subset of reachable MSM/FSW population. The double-counting and frequency adjusted estimate is a reasonable estimate for the currently active, visible and venue-based segment of the key populations "reachable" by intervention programmes. The mapping estimate, corrected for all three factors (double-counting, frequency and invisibility) represents an estimate of "all KPs". HIV prevention programme can also reach some KPs who are not necessarily found in venues.

Notable differences were observed among FSWs between double counting and frequency adjusted estimate ("reachable population") and previous estimates in the study areas, which may be explained by several factors. First, the difference in methods used—the consensus report was based on a Delphi method which involved imputation at the regional level while this study interviewed informants at the venues to generate the estimates in the wards that were directly mapped. The estimates from directly mapped wards were extrapolated to the regional level. Second, the consensus report is based on the knowledge and information as of 2014, while the mapping study was done two years later. It is likely that the business centres and venues may have changed and the FSWs may have migrated and increased in these areas due to the presence of pull factors. Third, the crude estimates were adjusted for correction factors to make the mapping size estimates into more appropriately usable size estimates, but this was not done in the consensus report.

Similarly, differences were observed between the 2013 IBBS study and estimates in the current study. Reasons for the discrepancy may include differences in the methods used (programmatic mapping size estimation vs. respondent-driven sampling). And it may also be likely that the scale-up of prevention programmes addressing stigma and discrimination in these areas done by Sauti and other previous projects may have created tolerance leading to more FSWs coming out to access services and disclosing their sexual activities. Studies have shown that higher utilisation of HIV services is positively correlated with key population's disclosure of their sexual activities and sexual orientations [34,35].

The programmatic mapping size estimation study was designed to get ward level data to assess the saturation of services in the Sauti project. Directly assessed mapping estimates were compared to programme reach data in the wards that participated in both the mapping size estimation study and Sauti project. Mapping size estimates are consistent to Sauti programmatic reach among FSWs in the study regions except for Iringa region where the double counting and frequency estimate of the FSWs was higher. Among MSM, the number reached by the Sauti project was higher compared to the number mapped in Mbeya and Njombe

regions while the mapping size estimation estimates in other regions were consistent to Sauti programmatic reach. The following factors may explain the differences and similarities in estimates between mapping study and Sauti project reach: First, different definitions used by the mapping study and Sauti project. The Sauti project used a very restrictive definition in that, FSWs were defined as female aged at least 18 years, who receive money, goods or favours in exchange for sexual services, as the primary source of income unlike a broader definition used by this study. Therefore, in areas where mapping estimates were consistent with the Sauti programmatic reach, mapping estimates were likely under-reported. The underreporting may be due to challenges of mobility in and out of the study areas, the hidden nature of FSWs due to legal, social and cultural factors. Second, the mapping size estimation study included MSM/FSWs accessing services and those who do not while KP programme data only describe KPs who access services at KP friendly facilities that disaggregate data by KPs. The use of KP-friendly facility data alone leaves an incomplete picture of KPs who access services in facilities where data is not disaggregated by sub-populations. Stigma continues to hinder uptake of services by KPs and disclosure resulting in low programme coverage and consequently, inadequate programmatic data for concluding the size of the population. It is also likely that Sauti project reached some KPs who are not necessarily found in venues by using social networks and other tools or programmatic data were not accurately deduplicated to individualise data. Data quality issues in routinely collected HIV service data are well documented [36–40].

The majority of FSWs and MSM in this study were based at the entertainment venues. This finding is consistent with the classification of FSW typologies reported in studies conducted in Kenya and Nigeria, where more than half of the FSWs were based in venues such as bars and nightclubs [20,24]. However, almost one-third of the MSM were not based at the venues. The use of mobile device applications has enabled MSM to get connected with other MSM efficiently at any time without necessarily visiting physical venues, especially when GPS application is activated. The continued rise in the number of smart-phone users and the frequent use of sex networking applications may continue to change the MSM networking in the coming years. This necessitates the need to conduct virtual mapping in addition to physical mapping to estimate MSM size, especially in settings where same-sex sexual activities are criminalised.

Several limitations of the geographic mapping size estimation should be acknowledged. First, this method is venue-based and therefore only identified venues mostly frequented by FSWs and MSM as reported by key informants. There is a possibility that some of the venues may have been missed and therefore under-estimated the number of FSWs and MSM. However, during level II, the existence of all venues reported by informants in level I was validated. Second, the method is based on key informant estimates rather than physical counting, which may lead to variability (over-estimation or under-estimation) between various respondents interviewed at the venue. The average of the estimates reported was derived and reported. This method could overestimate KPs if they frequented more than one venue during the mapping size estimation survey. To address this, the final estimates were adjusted for double-counting to correct for multiple counting. Third, the mapping size estimation study lacked good data for developing strata for extrapolation. Efforts were made to group wards with similar characteristics in the region into three strata (high, medium, low) based on rough estimates of the KPs in the ward estimated from previous projects and predisposing factors for convergence of HIV risk in the ward. In each stratum, wards were selected purposively to participate in the study. However, some of the wards stratified as low or medium tended to have a large number of KPs after mapping size estimation and vice versa. To mitigate the lack of good quality data for stratification in the pre-mapping phase, post-mapping stratification was done and used for extrapolation based on the observed values in the mapped wards and additional programmatic information in the unmapped wards (e.g. predisposing factors for HIV, the presence of pull

and push factors for KP in the ward). Extrapolation factors which use direct estimates in the numerator and general population census data for people aged 15–49 years in the denominator were used to generate extrapolated estimates. Dividing the direct estimates by the general population census data gives a "per capita" estimate for the ward, which is then applied to other wards. If there is a "mismatch" between the numerator (direct estimate) for the ward and the denominator (general population base) for the ward, the resulting extrapolation factor will be distorted. Moreover, this distortion will affect the extrapolated units in the stratum and ultimately will affect the overall estimates. Mismatches may occur when the numerator (direct estimate) counted people from outside the ward, denominator covered a much larger geographic area than the numerator or people were mobile and therefore counted in the numerator for multiple wards. Finally, the study used correction factors to make the mapping size estimates into more appropriately usable. The two adjustment factors (frequency of visiting sites and non-visibility) were making assumptions about people who may not have been at any venue during the mapping process. Therefore, the study collected information about MSM/ FSWs who may have been absent from MSM/FSWs who were present at the venue because they would be in a better position to inform those assumptions. On all three adjustment factors, the study did not ask the FSWs/MSM about their behaviour, but rather their views on other FSWs'/MSM client solicitation practices, including client solicitation at multiple venues. This approach may be prone to response bias, which could affect the associated assumptions built into the extrapolation models.

The key strengths of this mapping size estimation exercise lie in the effort to document the methods used in the study, data analysis strategies used, including key adjustments done. The aim was to provide estimates that can be used to improve planning, target setting for key population interventions, and assessing the coverage of the interventions provided to KPs. Another evident strength of the study was the involvement of the community (key population in the mapping size estimation survey). KPs participated in all stages of the study and therefore had a sense of ownership of the enumeration results.

## Conclusion

This study provides baseline figures for monitoring and evaluating HIV intervention services among FSWs and MSM in five regions of Tanzania. Ward, District and regional level data could be used for geographic prioritisation of the response by allocating more resources to areas with a large number of FSWs and MSM and program planning, target setting on who can be reached by peers or internet-based platforms and monitoring of the program coverage at the local level.

## Supporting information

**S1 Fig. FSW estimates by Category of Venue.**
(TIFF)

**S2 Fig. MSM estimates by Category of Venue.**
(TIFF)

## Acknowledgments

The authors would like to thank the Regional and District Health Authorities, Community gatekeepers, Civil Society Organisations, Key Population Groups, Sauti project management and the Mapping study team. We are also grateful to study participants for their acceptance to participate in the study.

## Author Contributions

**Conceptualization:** Mwita Wambura, Daniel Josiah Nyato, Neema Makyao, Mary Drake, Caterina Casalini, Soori Nnko, Gasper Mbita, Albert Komba, John Changalucha, Tobi Saidel.

**Data curation:** Jacqueline Materu.

**Formal analysis:** Mwita Wambura, Daniel Josiah Nyato, Evodius Kuringe, Tobi Saidel.

**Funding acquisition:** Mwita Wambura, Neema Makyao, Mary Drake, Albert Komba.

**Investigation:** Evodius Kuringe, Jacqueline Materu, Amani Shao.

**Methodology:** Mwita Wambura, Daniel Josiah Nyato, Evodius Kuringe, Soori Nnko, Amani Shao, John Changalucha, Tobi Saidel.

**Project administration:** Daniel Josiah Nyato.

**Supervision:** Mwita Wambura, Daniel Josiah Nyato, Mary Drake, Evodius Kuringe, Caterina Casalini, Soori Nnko, Gasper Mbita, Amani Shao, Albert Komba, John Changalucha, Tobi Saidel.

**Validation:** Jacqueline Materu.

**Writing – original draft:** Mwita Wambura, Daniel Josiah Nyato.

**Writing – review & editing:** Neema Makyao, Mary Drake, Evodius Kuringe, Caterina Casalini, Jacqueline Materu, Soori Nnko, Gasper Mbita, Amani Shao, Albert Komba, John Changalucha, Tobi Saidel.

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
