## [Decision Letter · Decision Letter 0]

10 Dec 2019

PONE-D-19-24277

Programmatic mapping and size estimation of key populations to inform HIV programming in Tanzania

PLOS ONE

Dear Dr Wambura,

Thank you for submitting your manuscript to PLOS ONE. After careful consideration, we feel that it has merit but does not fully meet PLOS ONE’s publication criteria as it currently stands. Therefore, we invite you to submit a revised version of the manuscript that addresses the points raised during the review process.

We would appreciate receiving your revised manuscript by 9th January 2020. To enhance the reproducibility of your results, we recommend that if applicable you deposit your laboratory protocols in protocols.io, where a protocol can be assigned its own identifier (DOI) such that it can be cited independently in the future. For instructions see: http://journals.plos.org/plosone/s/submission-guidelines#loc-laboratory-protocols

We look forward to receiving your revised manuscript.

Kind regards,

Kwasi Torpey, MD PhD MPH

Academic Editor

PLOS ONE

Journal Requirements:

**When submitting your revision, we need you to address these additional requirements:**

**Please ensure that your manuscript meets PLOS ONE's style requirements, including those for file naming. The PLOS ONE style templates can be found at http://www.plosone.org/attachments/PLOSOne_formatting_sample_main_body.pdf and http://www.plosone.org/attachments/PLOSOne_formatting_sample_title_authors_affiliations.pdf**

2. Thank you for including your ethics statement: "The geographical mapping study was granted ethics approval by the Medical Research Coordinating Committee (MRCC) of the National Institute for Medical Research (NIMR) in Tanzania (NIMR/HQ/R.8a/Vol.IX/2086), and Institutional Review Board of the Johns Hopkins University (IRB 00006668). Approval to work in the study communities was obtained through official permission from respective local government offices and leaders after authorisation from the regional and district government authorities. No personal identifying information was collected from respondents as part of the study. Interviewees provided verbal consent because sex work and same-sex relationships are illegal in Tanzania. The interviewers signed the consent form to confirm that they had obtained informed consent from the participant before conducting the interview."

a. Please amend your current ethics statement to include the full name of the ethics committee/institutional review board(s) that approved your specific study (the latter named approval for community interviews, specifically).

3. Please include additional information regarding the pretesting of the structured interviews used in the study, i.e. did pretesting take place and, if so, upon whom.

Reviewers' comments:

Reviewer's Responses to Questions

**Comments to the Author**

1. Is the manuscript technically sound, and do the data support the conclusions?

Reviewer #1: Yes

Reviewer #2: Yes

2. Has the statistical analysis been performed appropriately and rigorously? 

Reviewer #1: Yes

Reviewer #2: Yes

3. Have the authors made all data underlying the findings in their manuscript fully available?

Reviewer #1: Yes

Reviewer #2: Yes

4. Is the manuscript presented in an intelligible fashion and written in standard English?

Reviewer #1: Yes

Reviewer #2: Yes

5. Review Comments to the Author

Reviewer #1: This study, which sets out to map and estimate the number of female sex workers, men who have sex with men for the purpose of informing efforts directed at reducing HIV transmission in these hard to reach populations, is ambitious and in the main successful in identifying the places where these people can be accessed. Tanzania is large country geographically with diverse sociocultural populations. Tanga region, for example, is bordering Mombasa, a big seaport. It is not clear how it has influenced Tanga and might alter the picture that would emerge if it was included. There is therefore need to reflect this in the conclusions when discussing prevalence figures for Tanzania.

The design, methodology and conduct of the study has been technically and scientifically sound as well as of very high standard

Reviewer #2: The study addresses an important data and information need for effective HIV prevention programming in Tanzania. The study uses a methodology that has been applied in other countries in Africa and Asia for similar purposes. The methodology’s strengths and weaknesses are now well established. Overall, the study is described and presented well. The overall findings are also consistent with those from studies using a similar methodology in different parts of Africa.

The authors identify their three-step approach to adjust for potential double counting as the key strength of their study. However, I would advise that they provide further details on the adjustment factors, especially for double counting and frequency of visiting a venue. Reliable correction factors for key population (KP) size estimates from this methodology should be based on data on behaviours of KPs reached directly through the study itself, especially at Level 2 of the data collection process. The authors present on Table 1 (page 9) a set of questions that were used to derive adjustment factors for FSW estimates. The questions as phrased are not asking the FSWs about their own behaviour, but rather their views on other FSWs’ client solicitation practices, including client solicitation at multiple venues. Such an approach is prone to response biases, which could affect the associated assumptions built into the extrapolation models. Can the authors clarify if they asked the same set of questions to the KPs they interviewed focusing on the KP’s own behaviour and how the findings compared to the responses the KPs gave on their impressions of the practices of fellow/other KPs (as indicated on Table 1)? If not, can the authors comment on how not using data on actual behaviours of the sample of FSWs reached directly by the study could affect the assumptions for their population size estimation and extrapolation models?

6. PLOS authors have the option to publish the peer review history of their article (what does this mean?). If published, this will include your full peer review and any attached files.

Reviewer #1: No

Reviewer #2: Yes: Willis Omondi Odek

---

## [Author Response · Author response to Decision Letter 0]

15 Jan 2020

Reviewers' comments:

Reviewer's Responses to Questions

Comments to the Author

1. Is the manuscript technically sound, and do the data support the conclusions?

Reviewer #1: Yes

Reviewer #2: Yes

 2. Has the statistical analysis been performed appropriately and rigorously?

Reviewer #1: Yes

Reviewer #2: Yes

 3. Have the authors made all data underlying the findings in their manuscript fully available?

Reviewer #1: Yes

Reviewer #2: Yes

4. Is the manuscript presented in an intelligible fashion and written in standard English?

Reviewer #1: Yes

Reviewer #2: Yes

5. Review Comments to the Author

Reviewer #1: This study, which sets out to map and estimate the number of female sex workers, men who have sex with men for the purpose of informing efforts directed at reducing HIV transmission in these hard to reach populations, is ambitious and in the main successful in identifying the places where these people can be accessed. Tanzania is large country geographically with diverse sociocultural populations. Tanga region, for example, is bordering Mombasa, a big seaport. It is not clear how it has influenced Tanga and might alter the picture that would emerge if it was included. There is therefore need to reflect this in the conclusions when discussing prevalence figures for Tanzania.

The design, methodology and conduct of the study has been technically and scientifically sound as well as of very high standard

Response: Thanks for the comment. The study was conducted in regions supported by Sauti project to provide programmatic data for local program planning, target setting and monitoring of program coverage. Please see the last paragraph on page 4, line number 89-96. Extrapolation was used for estimating the size of FSWs/MSM in the wards without direct estimates within the study regions. This study was not able to generate national FSW/MSM estimates because only 5 regions were mapped. Extrapolating estimates in non-mapped regions using estimates in the five mapped regions may have generated imprecise estimates because of the lack of useful data to group regions into strata that are more likely to have similar population proportions of MSM/FSWs. 

Reviewer #2: The study addresses an important data and information need for effective HIV prevention programming in Tanzania. The study uses a methodology that has been applied in other countries in Africa and Asia for similar purposes. The methodology’s strengths and weaknesses are now well established. Overall, the study is described and presented well. The overall findings are also consistent with those from studies using a similar methodology in different parts of Africa.

The authors identify their three-step approach to adjust for potential double counting as the key strength of their study. However, I would advise that they provide further details on the adjustment factors, especially for double counting and frequency of visiting a venue. Reliable correction factors for key population (KP) size estimates from this methodology should be based on data on behaviours of KPs reached directly through the study itself, especially at Level 2 of the data collection process. The authors present on Table 1 (page 9) a set of questions that were used to derive adjustment factors for FSW estimates. The questions as phrased are not asking the FSWs about their own behaviour, but rather their views on other FSWs’ client solicitation practices, including client solicitation at multiple venues. Such an approach is prone to response biases, which could affect the associated assumptions built into the extrapolation models. Can the authors clarify if they asked the same set of questions to the KPs they interviewed focusing on the KP’s own behaviour and how the findings compared to the responses the KPs gave on their impressions of the practices of fellow/other KPs (as indicated on Table 1)? If not, can the authors comment on how not using data on actual behaviours of the sample of FSWs reached directly by the study could affect the assumptions for their population size estimation and extrapolation models?

 Response: Thanks for this comment. Only one of the three adjustments was for double-counting. The other two adjustments (frequency of visiting sites and non-visibility) were making assumptions about people who may not have been at any venue during the mapping process. Therefore, the study was collecting information about people who may have been “absent” from people who were “present” because they would be in a better position to inform those assumptions than if we just made assumptions randomly. 

Regarding double-counting, the study did not ask the FSWs/MSM about their behaviour, but rather their views on other FSWs’/MSM client solicitation practices, including client solicitation at multiple venues. We agree that such an approach may be prone to response bias, which could affect the associated double-counting assumption built into the extrapolation models. We have, therefore, acknowledged this as a weakness. Please see the last sentence on page 33, line number 639-649.

---

## [Editor Report · Decision Letter 1]

22 Jan 2020

Programmatic mapping and size estimation of key populations to inform HIV programming in Tanzania

PONE-D-19-24277R1

Dear Dr. Wambura

We are pleased to inform you that your manuscript has been judged scientifically suitable for publication and will be formally accepted for publication once it complies with all outstanding technical requirements.

With kind regards,

Kwasi Torpey, MD PhD MPH

Academic Editor

PLOS ONE
---

## [Editor Report · Acceptance letter]

24 Jan 2020

PONE-D-19-24277R1 

Programmatic mapping and size estimation of key populations to inform HIV programming in Tanzania 

Dear Dr. Wambura:

I am pleased to inform you that your manuscript has been deemed suitable for publication in PLOS ONE. Congratulations! Your manuscript is now with our production department. 

With kind regards,

on behalf of

Professor Kwasi Torpey 

Academic Editor

PLOS ONE